# Experiences of multimorbidity in urban and rural Malawi: An interview study of burdens of treatment and lack of treatment

Edith F. Chikumbu[1☉], Christopher Bunn[1,2☉]*, Stephen Kasenda[1], Albert Dube[1],
Enita Phiri-Makwakwa[1], Bhautesh D. Jani[3], Modu Jobe[4], Sally Wyke[2],
Janet Seeley[5], Amelia C. Crampin[1,3], Frances S. Mair[3], on behalf of the MAfricaEE
Project[¶]

1 Social Sciences Team, Malawi Epidemiology and Intervention Research Unit, Lilongwe, Malawi, 2 College
of Social Sciences, University of Glasgow, Glasgow, Scotland, United Kingdom, 3 College of Medicine,
Veterinary and Life Sciences, University of Glasgow, Glasgow, Scotland, United Kingdom, 4 MRC Unit The
Gambia, London School of Hygiene and Tropical Medicine, Fajara, Banjul, The Gambia, 5 Faculty of Public
Health and Policy, London School of Hygiene and Tropical Medicine, London, United Kingdom

☉ These authors contributed equally to this work.
¶ Membership of the MAfricaEE Project members are listed in the Acknowledgments.
* christopher.bunn@glasgow.ac.uk

pgph.0000139

Universitat Bremen, GERMANY

**Data Availability Statement:** All anonymised data
relevant to the study are available from the Malawi
Epidemiology and Intervention Research Unit's

## Abstract

Multimorbidity (presence of ≥2 long term conditions (LTCs)) is a growing global health chal-
lenge, yet we know little about the experiences of those living with multimorbidity in low- and
middle-income countries (LMICs). We therefore explore: 1) experiences of men and women
living with multimorbidity in urban and rural Malawi including their experiences of burden of
treatment and 2) examine the utility of Normalization Process Theory (NPT) and Burden of
Treatment Theory (BOTT) for structuring analytical accounts of these experiences. We con-
ducted in depth, semi-structured interviews with 32 people in rural (*n* = 16) and urban set-
tings (*n* = 16); 16 males, 16 females; 15 under 50 years; and 17 over 50 years. Data were
analysed thematically and then conceptualised through the lens of NPT and BOTT. Key ele-
ments of burden of treatment identified included: coming to terms with and gaining an under-
standing of life with multimorbidity; dealing with resulting disruptions to family life; the work
of seeking family and community support; navigating healthcare systems; coping with lack
of continuity of care; enacting self-management advice; negotiating medical advice; apprais-
ing treatments; and importantly, dealing with the burden of *lack* of treatments/services.
Poverty and inadequate healthcare provision constrained capacity to deal with treatment
burden while supportive social and community networks were important enabling features.
Greater access to health information/education would lessen treatment burden as would
better resourced healthcare systems and improved standards of living. Our work demon-
strates the utility of NPT and BOTT for aiding conceptualisation of treatment burden issues
in LMICs but our findings highlight that 'lack' of access to treatments or services is an impor-
tant additional burden which must be integrated in accounts of treatment burden in LMICs.

Data Documentalist on reasonable request.
Please contact Dr. Chifundo Kanjala Chifundo.
Kanjala@LSHTM.ac.uk.

**Funding:** This work was funded by an MRC Grant
awarded to FSM (MR/T037849/1). This study was
also supported by an allocation of the Global
Challenges Research Fund from the Scottish
Funding Council through funding awarded to ACC
(SFC/AN/10/2018). The funders had no role in
study design, data collection and analysis, decision
to publish, or preparation of the manuscript.

**Competing interests:** The authors have no
competing interests to declare.

## Introduction

Multimorbidity, the co-occurrence of $\geqq2$ long term conditions (LTCs) is a pressing global health problem [1], particularly in low- and middle-income countries (LMICs) [2]. Rising prevalence of chronic non-communicable diseases (NCDs) in LMIC is attributed to increasing urbanization, lifestyle changes, and advances in HIV treatment; [3] about 80% of global NCD-related deaths occur in LMICs [4]. In sub-Saharan Africa (SSA), NCDs are experienced along-side infectious diseases such as HIV and Tuberculosis (TB) [5]. Deaths from NCDs in Africa are projected to overtake infectious diseases, maternal, perinatal and nutrition-related deaths by 2030 [6].

A recent scoping review of the epidemiology of multimorbidity in LMICs [2] showed multi-morbidity is associated with older age [7–10] and female sex [7, 9, 11–13]. In high-income countries multimorbidity reduces quality of life, ability to be economically active, makes everyday tasks a continual struggle, and increases healthcare utilization and costs [14–17]. In LMICs, research has identified common challenges due to delays to care; access to medicines; increased costs as a result of referrals to tertiary care; and barriers to attending referral hospitals [18, 19].

Less is known about the experience of living with multimorbidity in SSA. Experiences of living with HIV and type 2 diabetes in South Africa have been explored [20] using a model of 'Cumulative Complexity' [21], which suggests that care of, and outcomes for, chronic illness are a result of a balance between workload demands (e.g. daily tasks of self-care, managing treatments, caring responsibilities and household tasks) and capacity to manage that workload (the extent of impairments, social and economic resources to help undertake the tasks and literacy). Patient workload was reported as high because of the time and effort required to attend multiple clinics, manage strict dietary requirements; pill burden; and the work of managing stigma. Capacity to manage this workload was reported as depending on a positive attitude, health literacy, family support, clinic support and availability of finances for recommended diets [20]. The paucity of comparable empirical work was noted.

In Malawi, the focus of this paper, epidemiology on multimorbidity is scarce. A single study offers a limited view on multimorbidity by assessing the prevalence of two or more of obesity, hypertension and diabetes in rural and urban settings, finding prevalence ranging from 1% in rural men to 7% in urban women [11]. These estimates are undoubtedly conservative, given high prevalence of HIV in these communities. Treatment for those living with chronic conditions in Malawi is delivered through government facilities which are free at point of delivery, but limited in provision [22], the Christian Health Association of Malawi and a range of private clinics, research organisations and non-governmental organisations.

Drawing on research in Malawi, we set out to contribute to the literature on treatment burden and experiences of multimorbidity in SSA through a theoretically informed account of living with multimorbidity, drawing on Normalization Process Theory (NPT) [23] and Burden of Treatment Theory (BoTT) [24]. NPT describes four aspects of treatment burden each of which involve work for the person with LTCs and which have been explored in a range of studies [24–26]: coherence (how people make sense of their condition(s) and treatment(s)); cognitive participation (how they interact with others to promote management of their condition); collective action (how they work individually and collaboratively to self-manage their condition in the context of their everyday lives) and reflexive monitoring (how they reflect on their efforts to care for themselves and adjust what they do in response).

BoTT incorporates and builds on NPT and cumulative complexity theory to take account of the factors that influence a person's ability to self-manage and cope with any given level of treatment burden [24]. It offers a series of 'generative principles' for understanding the

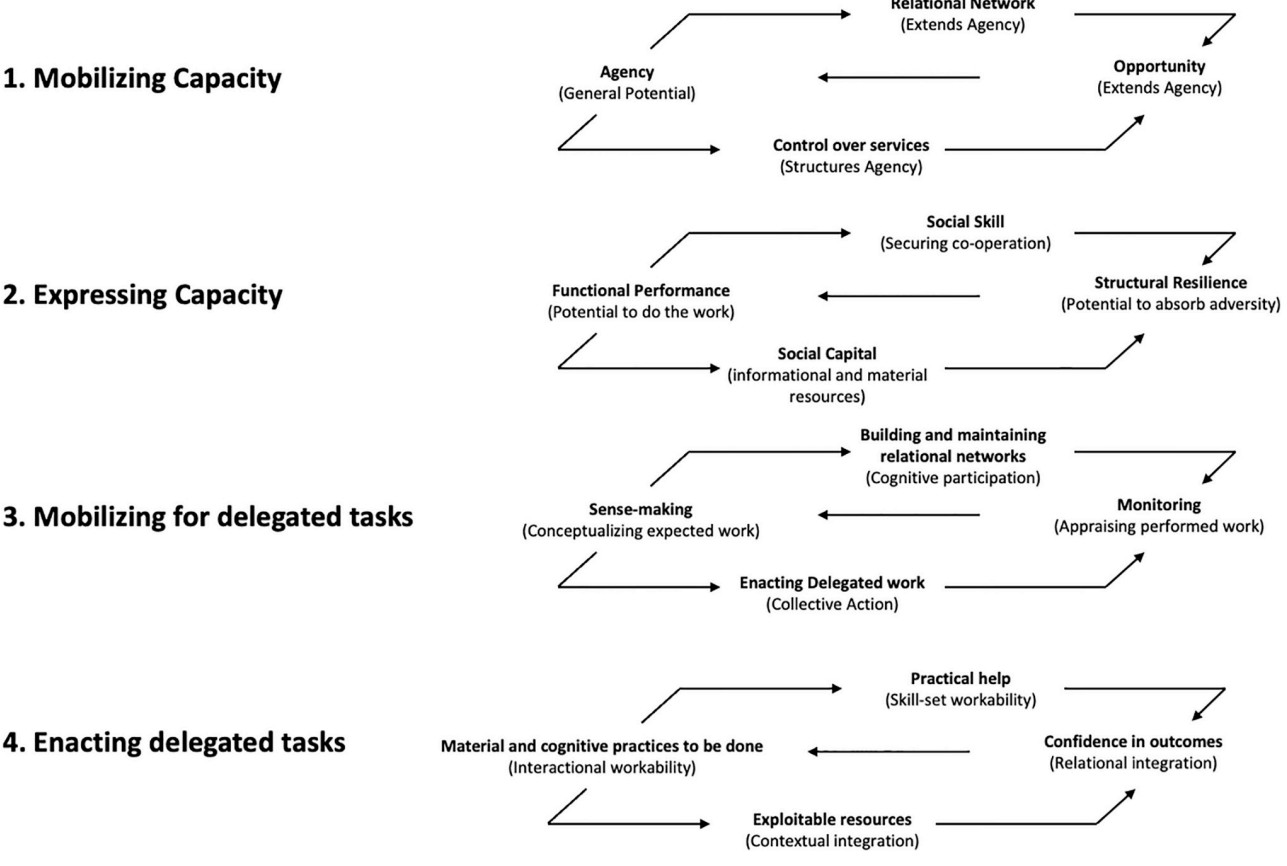

**Fig 1. Conceptual map of burden of treatment theory adapted from May et al. 2014.**

complex relational dynamics of the work of being a patient, the capacities patients have to manage their LTCs and interactions with healthcare systems. May et al. [24], who proposed BoTT, position patients as agents with capacities to manage LTCs which are potentiated or constrained by their social networks, controlled by the ways in which healthcare is organised and the opportunities accessible, available healthcare services offer. Patient capacities are seen as closely related to their functional performances–their bodily, material and cognitive capabilities–which are, again, potentiated or constrained by people in their social networks who provide support or whose actions inhibit capacity to self-manage. As patients work to care for their conditions, they utilise their capacities by making sense of the tasks they need to do to manage them, drawing on their social networks, enacting the tasks and then reflecting on the work they (and others) have performed. As well as offering an analytical approach to evaluating patient/healthcare interactions, May et al. sets out recommendations for interventions to improve health outcomes and patient experience, arguing that to improve patient capacity to undertake work requires strengthening collective capacity, social networks, and social capital [24]. We summarise BoTT in Fig 1.

NPT and BoTT seem to have the potential to support a generalizable understanding of multimorbidity in LMICs such as Malawi, but were developed in high-income settings and may not be either appropriate or relevant to LMICs such as Malawi, in that observed challenges for patients and their support community may differ at the system or individual level and the same may be true for factors that influence capacity. In this paper we aim to investigate: 1)

how those living with multimorbidity in urban and rural Malawi describe their experiences of life with multimorbidity and particularly their experiences of burden of treatment; and 2) how useful NPT and BOTT are for structuring analytical accounts of these experiences, and whether these theoretical frameworks need any refinements or adaptation to increase their utility in a LMIC context.

## Methods

We used semi-structured in-depth interviews to generate rich narratives of participants' experiences of living with multimorbidity which allowed us to better understand their experiences of burden of treatment. An interpretivist narrative approach was taken to enable participants and the interviewers to re-construct their journeys from diagnoses to present-day practices and experiences relating to life lived with multiple conditions. This approach has proven invaluable in past research on chronic illness and we sought to generate narratives of a similar nature and depth [27].

### Study setting, sample and recruitment

Participants with ≥2LTCs (diabetes, hypertension and HIV) who had granted permission for future contact were identified through searches of a database from a previous survey [11] and were recruited in Lilongwe and Karonga. Lilongwe is the capital city of Malawi and an important economic and transportation hub. Karonga is a rural district of Northern Region of Malawi where livelihoods are based on smallholder farming and fishing. This allowed comparison of experiences by area of settlement. As experiences of multi-morbidity may also differ by sex and age, purposive sampling sought to include males and females, and those aged both 50 and under and over 50.

Potential participants were telephoned and asked if they would accept a visit from a researcher who would like to introduce a new study to them. If they agreed, a researcher visited the home of the potential participant and presented the study in in participants' everyday language vernacular language (Chichewa or Chitumbuka), using information sheets (also in the vernacular) and encouraged participants to ask questions. A vernacular consent form was then read through with the participant and discussed. Written consent or thumb-print consent witnessed by a person of the participant's choosing was given by all participants.

### Data collection

After giving informed consent, participants were interviewed by a Malawian researcher in Chichewa or Chitumbuka to ensure the interview was conducted in participants' everyday language and that this language was also spoken by the interviewer. The topic guide (S1 File) asked about the impact of multimorbidity on lives and routines, as well as support from family and community. Interviews were audio-recorded using digital recorders, stored and transferred using encrypted devices, transcribed verbatim by transcription assistants whose first language matched that of each interviewee, translated into English and anonymised for analysis. Researchers and interviewers held regular meetings to discuss responses to the topic guide and evolving study findings. No changes to the topic guide were made during data collection. Interviews began in May 2019 and were completed in August 2019.

### Data analysis

To understand how those living with multimorbidity in urban and rural Malawi describe their experiences, data were analysed thematically [28]. Researchers (EC, CB) familiarised

themselves with a set of 8 transcripts, noting down impressions separately and generating initial codes which focused on the experience of multimorbidity. Secondly, they agreed on a coding frame which was applied to the full dataset. Thirdly, when the full dataset was coded, the researchers discussed how the coded data could be gathered under themes and then named the themes. Finally, the crosstab and matrix functions of NVivo 12 were used to allow exploration of patterns of difference in the thematic analysis by age, sex, and setting.

To consider how useful NPT and BOTT are for understanding accounts of multimorbidity in Malawi, we first mapped themes onto a previously described burden of treatment framework using four NPT constructs [24, 25] namely: making sense of multimorbidity (coherence); interacting with others (cognitive participation); enacting management strategies (collective action); and reflecting on management (reflexive monitoring). We assessed whether each theme was able to 'fit' one (or more) of the four domains or whether it fell outside the framework. Our analysis is presented using these four NPT domains. We also drew on the extended conceptual framework provided by BoTT to understand what influenced capacity to cope with any given level of treatment burden, to summarise the findings, and to identify any potential gaps in the theory. By producing an inductive thematic analysis of the data before mapping it onto the domains of NPT and BoTT, we sought to reduce the potential confirmation bias that would have arisen had we coded directly to the framework.

## Reflexivity

The researchers who conducted interviews were members of the communities in which the research took place. We believe that this enabled the building of rapport and trust, ensured culturally-specific references and framings were communicated and understood and provided interviewees with a familiar point of contact to support accountability. However, working with community researchers in the rural site may have introduced degrees of self-censorship in participant responses, for fear of potential breaches of confidence. We made this choice knowingly, calculating that the benefits of cultural familiarity outweighed the potential for participants to be overly guarded. The rich nature of the narratives we collected suggest this choice was justified. These concerns were not as great for the urban site, due to the significant size of the population.

The primary (EFC) and secondary (CB) analysts occupy very different social positions to both the interviewers and the interviewees, having access to greater economic, cultural and social capital and substantially different life experiences [29]. These characteristics undoubtedly limited the extent to which the analysts could fully apprehend the embodied experiences narrated by participants. However, both analysts have significant experience of working in these communities as researchers and, in EFC's case, as a clinician. These experiences include observing and participating in clinics and consultations which interviewees spoke about, as well as having family members living with multimorbidity who seek care in the same system as the interviewees. These experiences helped both analysts achieve degrees of empathy and insight that their social positions would otherwise have limited.

## Ethical review and community consultation

The study and its procedures were reviewed and approved by the National Committee on Research in the Social Sciences and Humanities in Malawi (ref: P.02/19/350); and College of Social Sciences Ethics Committee, University of Glasgow (ref: 400180124). Leaders and members of the two communities in which the study took place supported the research and welcomed the opportunity for the experiences of community members living with multimorbidity

to be amplified by researchers, expressing hope that the research would lead to improved healthcare in the future.

## Results

We recruited 32 people living with ≥2 LTCs, equally divided between the study sites and sexes, with 17 being at least 50 years old. Interviews lasted 25–144 minutes. The most common combination of conditions in the study sample was hypertension and type 2 diabetes (16), followed by hypertension and HIV (10), type 2 diabetes and HIV (4), and a combination of all three conditions (2).

Our thematic analysis produced 11 themes, 10 of which we judged to fit within the NPT framework (Table 1). The theme that we felt unable to allocate to the framework related to different kinds of 'lack' which our participants described. In presenting the analysis under the four NPT-related constructs we elaborate further consistency with key constructs in BoTT and summarise findings using a BoTT-based taxonomy in Table 2. Except where mentioned in the findings below, there were no differences between those living in different areas, different ages or between men and women.

### Making sense of multimorbidity (coherence)

In response to initial questions about how their LTCs started, participants offered accounts we organised as a theme *coming to terms with and reflecting on life with multiple LTCs*, consistent with the NPT theme coherence. Many of these accounts began with narratives that showed how participants explained the causes of their multiple LTCs. For example, a woman whose LTCs were identified at around the same time her child became sick saw that as the main explanation:

> In [year] my child died suddenly without getting sick. He was in [Secondary] Form 3, one day he came back from school and complained about headache. In the afternoon we went to [location] clinic where they referred us to [location] central hospital and he died on the same night of Monday. So, the anxiety that I had was what caused diseases because I was worried most of the times. I was asking myself about what happened and by living in such a situation, things were somehow not going well for me, so this created problems, and my blood pressure was just getting higher and higher. After diagnosing me with hypertension, they also diagnosed me with diabetes.

[Female LL ≥50 (DM+HTN)]

**Table 1. Themes constructed during thematic analysis mapped to four domains of treatment burden, adapted from Gallacher et al., 2013.**

| Treatment burden domain | Themes |
|---|---|
| Making sense of multi-morbidity | Coming to terms with and reflecting on life with multiple conditions |
| | Disruptions to life |
| Engaging with others | Family assistance |
| | The community and workplace |
| | Navigating the healthcare system |
| Enacting management strategies | Caring for the self |
| | Negotiating medical advice |
| Reflecting on management | Feedback on treatment |
| | Choosing care providers |
| | Suggestions for improvements |
| Unallocated | Lack |

**Table 2. Study findings mapped to BoTT.**

| **1: Mobilising capacity** | |
| --- | --- |
| 1.1: Agency | Multimorbidity limits agency through loss of hope, uncertainty and physical restrictions, in turn limiting capacity |
| 1.2: Relational Network | Relational networks could be enabling or constraining. They could increase capacity to manage LTCs by providing a range of supports (psychological and practical) but alternatively could have negative effects (e.g. forcing people to travel to escape the gaze of community members and potential stigmatisation). |
| 1.3: Opportunity offered by health services | Significant absences of opportunity limited patient capacity. Specifically, poor access to medicines and specialist healthcare providers |
| 1.4: Control—service organisation | Variability of care provided by healthcare providers and frequent changes in providers limit capacity. |
| **2: Expressing capacity** | |
| 2.1: Social Skill | Those able to enlist others in their care were able to increase their capacity (e.g. through convincing family to support self-care needs). |
| 2.2: Functional Performance | Loss of function (e.g. amputation) led to significant restrictions in capacity to self-care. Health literacy and cognitive abilities also are significant mediators of capacity. Regaining functionality via treatment produced increases in patient capacity. |
| 2.3: Structural resilience | Resilience was derived from local community groups, family and work colleagues, increasing capacity. Stigmatising discourse in communities and workplaces limits resilience, decreasing capacity. |
| 2.4: Social Capital | Social connections provided some with access to information (e.g. how to use a glucometer) and material resources (e.g. small contributions from local community groups). |
| **3: Mobilising for delegated tasks** | |
| 3.1: Sense Making (coherence) | Capacity to make sense of self-care tasks is limited by how and what instructions are given (or not) on diagnosis. Sense-making could be limited by entanglement with trauma. |
| 3.2: Building and maintaining relational networks (cognitive participation) | Rural participants found it easier to build and maintain relational networks with healthcare providers than urban participants, due to greater stability in healthcare provider staffing resulting in increased continuity of care/trust. |
| 3.3: Enacting delegated work (collective action) | |
| 3.31: Material and cognitive practices to be done (interactional workability) | Changes in diet were supported where families enabled them, enhancing capacity to self-care. Family members could also limit capacity (e.g. husbands reacting badly to condoms). |
| 3.33: Practical Help (skill set workability) | Assistance with food preparation increases capacity to enact dietary change. The absence of practical assistance for those working away from home limits capacity to self-care. |
| 3.34: Exploitable resources (contextual integration) | Those with access to economic resources could enhance their capacity for self-care (e.g. by purchasing medicines unavailable at government clinics/devices to support treatment), unlike those without economic resources. |
| 3.35: Confidence in outcomes (relational integration) | Those experiencing positive outcomes from treatment improved capacity to self-care. Those without confidence to pursue an outcome (e.g. using condoms, faced reduced capacity for self-care). Confidence in the skills of health professionals affected trust and willingness to engage with health services and follow advice. |
| **4: Monitoring** | |
| 4.1: Reflexive monitoring | Noticing positive outcomes from treatment led to increased capacity to self-manage (e.g. cultivating food). While avoiding stigma, increased capacity by removing the negative impacts of stigmatisation, but required additional capacity to achieve (e.g. travelling further and enlisting support of family members). Standards of life limited people's capacity to achieve functional performance |

When describing life with multimorbidity some participants recounted adapting to the reduced capacities of their bodies, for example by leaving jobs which involved heavy labour and starting a small business instead. Others spoke of the impact of multimorbidity on how they manage the future. For example, one of the younger men explained how his LTCs limited his agency and capacity to act in the world (using the Chichewa term word *chopinga*, which can translate as 'hindrance' or 'challenge') consistent with BoTT's 'mobilising capacity (agency)' construct [24]:

> It is very painful for someone who is having incurable or chronic diseases because such person loses hope. . . what I mean is if you are suffering from chronic diseases you lose hope as if you are dying today or tomorrow and that becomes a hindrance since it's like you are waiting for something to happen
>
> [Male ≥50 (HTN + DM)]

Among the accounts of life with multimorbidity provided, *disruptions to life* featured frequently, consistent with the concept of 'functional performance' outlined in BoTT [24]. In some instances, these disruptions were very limiting:

> I was employed during that time, but after they amputated my leg at the hospital, I wouldn't have managed to go to work with a walking stick. I was doing work of welding and fabricating and I was supposed to be climbing at the top of buildings, it was challenging for me. So I just decided to go and sit down home.
>
> [Male LL ≥50 (HTN+DM)]

Our data suggests that participants' attempts to make sense of multimorbidity were focused on establishing causal narratives, the impacts multimorbidity has on their capacity to act in the world, their functional abilities, and on the disruptions it has brought to their lives.

### Engaging with others (cognitive participation)

From questions about how others helped or did not help with managing participants' multimorbidity we identified three themes which are consistent with the NPT construct cognitive participation: *family assistance*, *interactions in the workplace*, and *navigating the healthcare system*, which all shaped capacity to carry out the work of life with multimorbidity.

The management of the three LTCs included in this study demands adjustments to lifestyle, with diet being particularly important for type 2 diabetes and hypertension. One participant described how to avoid spending more money and time in preparing separate meals his whole family adopted the diet he had been told to adopt to help manage his diabetes. They used *mgaiwa*, a whole grain maize flour instead of *ufa woyera* a de-husked and refined flour, to prepare the staple stiff porridge, *nsima*. Due to the differences in, the glycemic index, *mgaiwa* is recommended for diabetics. The family would prefer *ufa woyera*, making a sacrifice for the father to enhance his capacity to care for himself, consistent with the concept, taken from BoTT, of enacting delegated tasks, and the practical help offered in preparing the food [24].

Other forms of *family assistance* included providing money for transport to clinics, food supplies and money for medicines. Family members collected medicines too and reminded interviewees of their hospital appointments. Multiple accounts collected in our interviews described family members as a great source of encouragement and moral support. For example one of the younger women explained that her brothers travelled all the way from South Africa to encourage her, offering practical and emotional support, consistent with the line of

argument in BoTT [24] that enabling social capital (through family support) can enhance capacity to self care:

> I have two brothers, my brothers are living in Johannesburg [. . .] yes, so they also gave me an encouragement that, "Sister, you should be praying each and every day, you should know God, yes if you know God, He'll give you more days. Sicknesses goes but you should not be sorrowful. Yes, when they heard [they] came from Johannesburg they came to give me this advice.
>
> [Female <50 (HTN+DM)]

There was a difference in how women and men described the reactions of their communities and workplaces to their LTCs. Women's accounts focussed on the response of community members, demonstrating that social networks can enhance a person's capacity to self-manage their illnesses:

> Aaah, sometimes people come to see me. For instance, my fellow choir members come with money, Home Based Care women also come. I also belong to other groups like Tikondane, Ladies Macheza, so they also come to see me. In addition to that, our community also come to see me. When coming to see me, they don't come empty handed, they come with a little something.
>
> [Female <50 (HIV+HTN)]

Other women spoke about how the fear of the continued stigma faced by those living with HIV led to them keeping their diagnoses to themselves, suggesting how stigma can inhibit 'building and maintaining relational networks' identified as a key process enabling self-care in the BoTT framework [24].

Men's accounts, on the other hand, focussed on response in the workplace. For example, stigma was also present in the account that one younger man gave of the mixed reactions he received from co-workers when he disclosed that he had diabetes:

> They encouraged me to follow advice from the hospital which told me because if I follow the advice, I might live longer and not get sick frequently [. . .] Others were speaking in a mockery way saying this person suffers from diabetes and saying some words. I was worried with those words and would ask myself how come [. . .] my friends are talking about me and mocking me.
>
> [Male KR <50 (HTN_DM)]

Another described how his workplace routines made it challenging for him to stick to his medication plan, particularly when working on jobs away from home. The workspace was a challenging place for younger men living with multimorbidity: some 'structural resilience' and 'social capital' is available to enhance capacity, but they are also subject to mockery and to unpredictable working patterns which present challenges for the 'workability' of self-care, reducing capacity [23, 24].

When participants described *navigating the healthcare system* a range of issues were discussed. A primary concern was waiting times:

> The other thing which discouraged me at [clinic] was; if you go there, you should just dedicate the whole day without having any programmes because it takes time to meet the doctor
>
> [Male LL <50 (HTN+DM)]

The disruption to the day that substantial waiting times cause was more commonly noted by younger and urban interviewees. These interviewees were of working age and had work, business or caring needs that they had to balance with their need for healthcare, compromising the 'workability' of interacting with the healthcare system [24]. The under-staffed and poorly-resourced health systems with which participants were trying to interact limited opportunities, increased treatment burden and diminished capacities to undertake the work of self-care [21].

Participants also spoke of many and routine shortages of medicines in the public health system. The shortages can also be understood of the lack of opportunity of health systems to support capacity to self-manage. While an urban male interviewee could resort to trying to buy medications from local pharmacies, a rural female interviewee who needed insulin injections could not get her medicine in the absence of public provision, significantly limiting her capacity to care for her diabetes.

Participants also spoke of seeing different staff each time they attended clinic. As one participant described:

> They have different doctors in the rooms. Sometimes I go there this month and meet a certain doctor. When I go there the next month, I also meet another doctor. It takes time for me to meet the doctor.
>
> [Female LL <50 (HIV+HTN)]

The lack of continuity of care might constrain patient capacity, with the interactional workability of consultations limited by social ritual or clinical details withheld due to a failure to establish trust [24]. Such accounts were more prevalent among urban participants. In the rural setting, few mentioned changes to the medical personnel they saw and when they did, it was not a topic of concern ultimately leading to higher levels of confidence in outcomes [24].

Interactions with family, community, work colleagues and healthcare providers play important roles in supporting or limiting the capacity of those living with multimorbidity to carry out the work of self-care. Through some of these relationships, substantial material and emotional support is obtained. Through others, stigma is applied, time is wasted, medicines are absent, and trust is limited by staff turnover.

## Enacting management strategies (collective action)

From questions about how participants managed their conditions we identified two themes: *caring for the self* and *negotiating medical advice* consistent with the NPT theme collective action. Many participants described in great detail the measures they took and challenges they faced in order to stay well.

Dietary changes were discussed extensively. For those living with hypertension, the need to reduce salt intake presented a challenge:

> It took a long time. Sometimes when I add little salt to relish, nsima was not tasting well for me, so I was adding extra salt in my plate when eating. When I started reducing salt, I was able to feel that there is a change in how I used to stay compared to the previous time. [. . .] As of now when I measure my blood pressure, they find it normal.
>
> [Female LL <50 (HIV+HTN)]

This account, echoed by others highlighted how enacting dietary change could be a struggle but in this instance it was validated by an embodied feeling of positive change, which was subsequently backed up by clinical blood pressure measurements.

Other participants described caring for type 2 diabetes using a glucometer. One man described how he used the device:

> I bought a testing device which I use to check myself every day. During the time I was employed, it was helping me because I was walking a long distance to minibus depot which was part of exercise and with the nature of my work, I was working under the sun and I was sweating, so it was also part of my physical exercise and when I test my sugar the following morning it was at a good level such as 90.
>
> [Male <50 (HTN+DM)]

Following BoTT, this man demonstrated that he had the mental and physical capacity (functional performance) and financial resources (exploitable resources) to enable him to purchase both the glucometer and testing strips, combined with his socially acquired understanding of how to use it ('social capital' and 'social skill') enabling him to cope with adversity (loss of work), and sustain optimal blood glucose levels [24].

While these two examples of *caring for the self* present positive pictures of how our participants approached the work of self-care, this was not the case for all. A common response to life with multimorbidity, particularly from urban participants, was to express frustration with medications. The daily work of consuming multiple medications was recognised as 'tiresome work' and some expressed extreme reluctance to accept this workload for life, pointing to a rejection of what in the BoTT framework is seen as a 'workability' of 'delegated tasks' [24].

During the interviews, participants described *negotiating medical advice*. Specifically, the ways in which they critically appraise the self-management strategies they are asked to enact in relation to their life situations and priorities. For example, one woman who was diagnosed with HIV and hypertension while pregnant reframed the tasks she was asked to carry out in relation to her priorities, embodied feeling and assessment of whether tasks were feasible (considered their 'interactional workability' in the BoTT framework):

> [. . .] they advised me that I should start taking medicine. They also advised me that I should be using condoms during sexual intercourse. So, there were many things that they advised me [. . .] They advised me this, but because I went there to attend antenatal clinic and I was also feeling well, I considered that as a useless thing [. . .] I just said aaah, with the behaviour of my husband how will I initiate this? I decided that I should first deliver my baby.
>
> [Female LL <50 (HIV+HTN)]

Other participants described how they made sense of and responded to dietary recommendations that they received from clinicians. For example, for the man in this extract, retaining and understanding the advice he received was not a problem, but he felt that financial constraints rendered enacting the advice impossible (exploitable resources):

> I was advised to (as for sugar) stop taking cooked half processed maize, bananas, some sweet things like oranges, and if I may decide to take oranges I should take one, not proceeding two and I should not take bananas excessively. Yes, but I saw that due to my poverty then I just eat without selecting (laughs)
>
> [Male KR <50 (DM+HIV)]

Some accounts of enacting self-management strategies described how careful and considered their approaches to *caring for the self* were. Others noted the burdensome nature of life

with multimorbidity, particularly in relation to polypharmacy. We also found that participants *negotiated medical advice* in relation to the resources available to them and their assessments of priority and likely outcomes.

### Reflecting on management (NPT reflexive monitoring)

As we have seen, some participants in this study appraise the work they do to care for their LTCs and make decisions based on this. We identified three themes which together are consistent with the NPT theme reflexive monitoring: *reflections on treatment*, *choosing care providers* and *suggestions for improvements*.

Participants commonly linked the introduction of treatment with loss of symptoms and increased functionality:

> Before I started taking my medication my heart could beat faster. In the case of diabetes, before I had knowledge of it I couldn't cultivate because I felt very weak and I couldn't figure it out that it was due to diabetes, but after I started taking medication now I can cultivate for long time [. . .] unlike before.
>
> [Female KR ≥50 (DM+HTN)]

Thus, consistent with BoTT, we can see that successful treatment of multimorbidity increases patient capacity, especially through improvements in physical wellbeing (functional performance), enabling them to continue to enact self-management strategies, grounded in confidence in the outcomes they have experienced [24].

A less common reflection on management related to *choosing care providers* and is closely linked to the stigma experienced by some participants when interacting with others. Being seen attending a clinic was a focal concern:

> Many people go to those hospitals because they regard them as convenient hospitals. I can lack peace if I can meet with people who know me [other community members]. They can start telling other people that I take these medicines [ARVs], so this can really concern me. I manage to go there [a different clinic]. I disclosed to my relatives so they can manage to go there, if I can become sick.
>
> [Female LL <50 (HIV+DM)]

This example demonstrates that participants can intentionally increase their burden of treatment to protect their social identities from the detriments of stigma, increasing the work of their social networks.

Finally, participants reflected on ways in which their treatments might be *improved*. One issue which was raised related to health education:

> We diabetic people tend to forget things easily, sometimes we can mess up things. There are other people who are recently introduced on medication, so they are supposed to be taught before they start getting medication. They should know about the foods which they are supposed to eat. As someone who was diagnosed long time ago, I can also forget things.
>
> [Female LL <50 (HTN_DM)]

Others spoke of receiving a 30 minute briefing at diagnosis, and then receiving no further informational input. In these ways, and in line with BoTT, participants highlighted the need for access to 'informational resources' to enhance their capacity for self-care [24].

A second group of *suggestions for improvement* related to material provision. These participants made suggestions such as:

> They [the government] can help us in terms of money because for somebody to have a better life you need to have money. Like small business, if the government can form a group for those people who have chronic illnesses and give them something for them to be doing to help them to improve their lives.
>
> [Female KR <50 (DM+HTN)]

The participants draw attention to the problem of poverty and the importance of improving standards of living (better housing, better food and money) to promote physical wellbeing.

## Lack

Across all themes we identified common narratives of absences or *lack* in relation to life with multimorbidity were present. It is clear from the data already presented that people living with multimorbidity in Malawi do face treatment burdens, but participants also carry the burden of *lack* of treatment. In this section we explore this theme, before returning to how it relates to BoTT in the discussion.

A focal *lack* reported by many was appropriate food, particularly among those living with diabetes. There was a perceived inability to avoid carbohydrate heavy, fatty and processed foods because these are the less expensive foods and are easily available. As one woman noted:

> There are other people who lose their life due to lack of assistance on their diabetes disease. The problem is that we fail to access things because of lack of money. Had it been that we have enough money, we could have been buying things that we are required to eat that time. The major issue about diabetes is on diet. This is what creates problem because we lack these foods.
>
> [Female LL <50 (HTN+DM)]

Another participant pointed to how *lack* of capacity in the health service directly shapes how he approaches his care:

> Sometimes you can spend much time there but when you go to [the clinic] pharmacy you could find that the medicine is out of stock and you are told to buy, so it was like being troubled very much and this discouraged me to go again. When I run out of medicine to go and be on a long line of patients, and I said "what if I just find money and buy medicine". This was easier to me because I just look for money, buy medicine and take them.
>
> [Male LL <50 (HTN+DM)]

This combination of lack of staff and lack of medication (lack of opportunity of healthcare services to support capacity) led this patient to pursue unsupported management of his conditions.

Finally, some participants perceived a lack of specialist knowledge among the clinicians who managed some of their conditions. For example, in this example the participant suggested

the lack of specialist care is a problem potentially affecting his confidence in the care offered which is consistent with BoTT in relation to confidence in outcomes (relational integration)):

> They [the government] should recognize diabetes as a disease and that many people are suffering from it. When posting doctors who are supposed to work in hospitals, they should consider sending doctors who are well trained on diabetes issues. So that they should be able to manage patients who come for assistance.
>
> [Male LL <50 (HTN+DM)]

Participants highlighted key absences in their treatment emphasising that "*lack*" can be an important issue in SSA. Participants linked diabetes mortality in their communities to lack of funds and supply of appropriate food; they reflect on a healthcare system's lack of capacity to meet their needs respond by self-treating without qualified supervision; and they perceive a lack of specialist care for a disease which is affecting growing numbers of their community.

## Discussion

These findings illuminate the multiple and intersecting aspects of burden of treatment which people living with multimorbidity in urban/rural Malawi experience in their daily lives.

We have shown that the key concepts of burden of treatment: gaining an understanding of life with multimorbidity (coherence); the engagement work of seeking family and community support (cognitive participation); enacting self-management advice (collective action); and appraising treatments (reflexive monitoring) identified through NPT and BOTT are described by people with multimorbidity in SSA. However, we have also shown that dealing with the burden of *lack* of treatments/services is a key issue.

Future work should therefore examine the processes through which this burden manifests in greater detail. We have shown a similar interplay between treatment burden and capacity issues in Malawi to that seen in HICs and that these issues can be usefully assessed and unpicked using both NPT and BoTT approaches, which highlight the social processes through which capacities and resources interact to produce greater or lesser experiences of burden. The work illustrates how poverty and inadequate healthcare and information provision in SSA constrained capacity to deal with treatment burden while supportive social and community networks were important enabling features.

Our findings resonate with other work in Malawi and SSA showing how people face challenges accessing care, encountered difficulty obtaining medicines from public hospitals and faced financial barriers to treatment, leading to delayed care and, most likely, poorer clinical outcomes [19]. Our findings reinforce research that suggests Malawi's 'Essential Health Package' (the minimum government provision) continues to experience important gaps in provision [22, 30]. Our findings also overlap with work in South Africa which identified burdens relating to a lack of holistic care which took account of all of a person's problems, the amount of time taken up by treatment, limited treatment choices and medication burden. Likewise, we also found that participants reported experiencing overcrowding in clinics, drug shortages and stigma [20]. Where our study goes beyond these contributions is in the accounts it offers of the sense-making work that our participants did to locate their multiple illnesses within their biographies [27], grapple with questions of identity and self [31, 32], and to narrate disruptions to their lives [33].

The generative principles of BoTT enabled us to bring out the links between patient capacity and workload, as mediated by social networks, healthcare systems and situated opportunities. It provided the conceptual toolkit to demonstrate how these domains structure action

through feedback loops, for example, when a patient's appraisal of long waiting times and drug shortages led to an un-supported approach to discharging the work of self-care. The theory also enabled us to highlight the importance of social networks and the resources which they can bring to lighten the burden of treatment, by adding to both capacity and resilience, as exemplified by the woman who received many community visitors who all brought her 'a little something'. Finally, the concept of 'functional performance' was particularly useful for illuminating the positive effect successful treatment of multimorbidity can have by re-enabling participants to contribute to their social networks (e.g. through cultivation), (re-)building social ties which enable future social support and thus greater resilience/capacity to manage adverse consequences of their multiple illnesses.

While Burden of Treatment Theory does include the concept of 'opportunity' which addresses the issue of availability of services in a given context, this concept incorporates impacts on the people's capacity to self-manage with a given level of treatment burden rather than explicitly identifying this as an additional cause of treatment burden. In addition, this concept of 'opportunity' does not address lack of access to medications which was a key issue identified here. Lack of access to good care is a burden *in and of itself*: to live in the knowledge that there are treatments and resources available that will improve your health, but that these are withheld from you for economic or spatial reasons [34, 35].

Importantly, none of the three measures of treatment burden developed thus far [36–38] explicitly measure lack of treatment as a component of burden of treatment. Indeed, there is an implicit assumption that medications and therapies are available, albeit perhaps difficult to access at times or in ways that suits people's needs [26]. We would therefore suggest that measures of treatment burden in LMIC contexts will need to be adapted to measure the burden of "lack of treatment" as a specific domain and barometer of quality of care. In addition, this concept may also be important when assessing burden of treatment with socioeconomically disadvantaged populations more widely, especially in contexts where there is not access to universal health care provision, free at point of care. We think of this as the *burden of lack of treatment*, of living with the knowledge of what is missing in your treatment and propose to integrate this with BoTT (see Fig 2).

As policy makers in Malawi and the wider SSA region contemplate how to respond to the growing burden of multimorbidity associated with the rapid rise of NCDs, much can be learnt from the experiences of those who live with multimorbidity. Their lives involve significant additional work to remain well and to contribute to their families, communities and wider social networks. In many instances, this work is shared with those in wider social networks that increase patient resilience and capacity, enabling better navigation of healthcare systems. The priorities for the participants in this study relate to: better access to high quality care including with less waiting time and reliable provision of medicines; improved standards of living (housing, food, income) to allow the resources through which people can better self-manage; and greater access to informational resources. Attending to these needs would substantially reduce both the burdens of treatment *and* of lack of treatment experienced by those living with multimorbidity in Malawi and possibly SSA more widely.

## Strengths and limitations

Our study has made a contribution to what is a very limited literature, making comparisons difficult [20]. However, our contribution builds on this work and has highlighted the potential of theory-engaged work to illuminate the complex lived experiences of those with multimorbidity. While our study included people living with three different conditions in a variety of combinations, a broader sample that encompassed a greater number of conditions would have

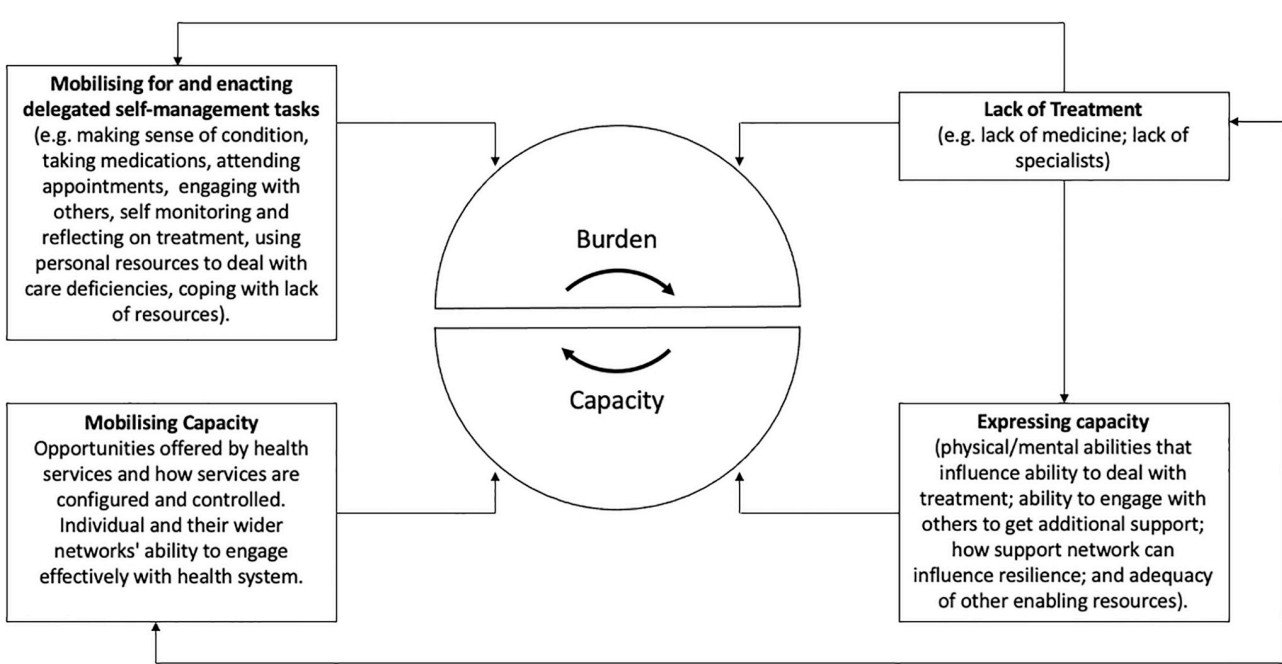

**Fig 2. Burden of treatment theory with the integration of the construct of lack of access to treatments or services.**

been desirable and would likely have given us insight into the implications of living with multi-morbidity that involves combinations of conditions which receive less attention in SSA e.g. arthritis, epilepsy. Finally, while many interviewees gave critical accounts of the care they receive, not all did; it is possible that some feared repercussions from healthcare professionals and other authorities or that respect for those in professional positions tempered criticism of difficult experience.

## Conclusions

Our study contributes to the emerging literature on experiences of multimorbidity in SSA and LMIC countries more broadly and demonstrates the utility of NPT and BOTT for aiding conceptualisation of treatment burden issues in LMICs. However, our findings highlight that 'lack' of access to treatments or services is an important additional dimension of the concept of burden of treatment which merits further investigation and should be integrated into future measures of treatment burden in SSA and likely other LMICs. Our work has shown that greater access to health information and education would lessen treatment burden as would better resourced healthcare systems that provide more continuity. Poverty also constrained capacity to deal with treatment burden suggesting the need for improved standards of living as a key issue. Nonetheless, supportive social and community networks were important enabling features and policies to support these networks would be important.

## Supporting information

**S1 Checklist. Standards for reporting qualitative research checklist.**
(DOCX)

**S1 File. Topic guide used in this interview study.**
(DOCX)

## Acknowledgments

The lead author would like to thank the participants for their time and the two ethics committees for their helpful reviews. The authors would like to acknowledge members of the wider MAfricaEE project team: Dr. Gertrude Chapotera, Prof. Andrew Prentice, Dr Alison Price.

## Author Contributions

**Conceptualization:** Christopher Bunn, Janet Seeley, Amelia C. Crampin, Frances S. Mair.

**Data curation:** Edith F. Chikumbu, Christopher Bunn, Amelia C. Crampin.

**Formal analysis:** Edith F. Chikumbu, Christopher Bunn, Sally Wyke, Frances S. Mair.

**Funding acquisition:** Christopher Bunn, Bhautesh D. Jani, Modu Jobe, Janet Seeley, Amelia C. Crampin.

**Methodology:** Christopher Bunn, Sally Wyke, Amelia C. Crampin, Frances S. Mair.

**Project administration:** Edith F. Chikumbu, Christopher Bunn, Stephen Kasenda, Albert Dube, Enita Phiri-Makwakwa, Amelia C. Crampin, Frances S. Mair.

**Resources:** Amelia C. Crampin.

**Supervision:** Christopher Bunn, Stephen Kasenda, Albert Dube, Enita Phiri-Makwakwa, Amelia C. Crampin, Frances S. Mair.

**Validation:** Edith F. Chikumbu, Christopher Bunn.

**Writing – original draft:** Edith F. Chikumbu, Christopher Bunn, Frances S. Mair.

**Writing – review & editing:** Edith F. Chikumbu, Christopher Bunn, Stephen Kasenda, Albert Dube, Enita Phiri-Makwakwa, Bhautesh D. Jani, Modu Jobe, Sally Wyke, Janet Seeley, Amelia C. Crampin, Frances S. Mair.

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
