## [Decision Letter · Decision Letter 0]

26 Oct 2021

PGPH-D-21-00403

Experiences of multimorbidity in urban and rural Malawi: an interview study of burdens of treatment and lack of treatment

Dear Dr. Bunn,

Thank you for submitting your manuscript to PLOS Global Public Health. After careful consideration, we feel that it has merit but does not fully meet PLOS Global Public Health’s publication criteria as it currently stands. Therefore, we invite you to submit a revised version of the manuscript that addresses the points raised during the review process.

Three reviewers have assessed your submission. Overall the views on your paper are very positive and I believe addressing the reviewer comments will further strengthen the manuscript. We therefore invite you to revise and resubmit your paper. You can find the reviewer comments below. Please let me know if you have any questions about the process.

We look forward to receiving your revised manuscript.

Kind regards,

Melanie Boeckmann

Academic Editor

Journal Requirements:

1. Please provide additional details regarding participant consent. In the ethics statement in the Methods and online submission information, please ensure that you have specified what type you obtained (for instance, written or verbal, and if verbal, how it was documented and witnessed). If your study included minors, state whether you obtained consent from parents or guardians.

2. In the online submission form, you indicated that "All anonymised data relevant to the study are available from the authors on reasonable request."

3. State what role the funders took in the study. If the funders had no role in your study, please state: “The funders had no role in study design, data collection and analysis, decision to publish, or preparation of the manuscript.”

Reviewers' comments:

Reviewer's Responses to Questions

**Comments to the Author**

1. Does this manuscript meet PLOS Global Public Health’s publication criteria? Is the manuscript technically sound, and do the data support the conclusions? The manuscript must describe methodologically and ethically rigorous research with conclusions that are appropriately drawn based on the data presented.

Reviewer #1: Yes

Reviewer #2: Yes

Reviewer #3: Yes

2. Has the statistical analysis been performed appropriately and rigorously?

Reviewer #1: Yes

Reviewer #2: N/A

Reviewer #3: N/A

3. Have the authors made all data underlying the findings in their manuscript fully available (please refer to the Data Availability Statement at the start of the manuscript PDF file)?

Reviewer #1: Yes

Reviewer #2: Yes

Reviewer #3: Yes

4. Is the manuscript presented in an intelligible fashion and written in standard English?

Reviewer #1: Yes

Reviewer #2: Yes

Reviewer #3: Yes

5. Review Comments to the Author

Reviewer #1: Thank you for submitting this manuscript to PLOS Global Health. The manuscript is well-written and reflects the quality of work that went into the project. It presents important findings that not only apply to the Malawian but wider SSA and LMIC contexts. Having read the paper, I have identified some minor comments, which I hope will improve the readability of the paper further:

Abstract: The abstract mentions “navigating health systems” twice.

In the abstract conclusion, it would be better to put the utility of NPT and BOTT after the main findings, in line with how the study was structured and presented.

Introduction: The introduction is well written overall and presents key figures for context. In some places, however, the authors have used a number of references to support a single statement (see lines 88-89 as an example). It would be worthwhile retaining the 1-2 most relevant or recent references in this situation.

In the opening paragraphs, Sub-Saharan Africa is mentioned however the authors quickly move to Malawi in the text that follows. It would be good to discuss the wider context before Malawi is brought into focus. Doing so, it is hoped, would provide a reasonable justification for conducting this work in Malawi.

Available evidence on multimorbidity in Malawi seems to suggest a very low burden, can authors provide a brief commentary on this (single sentence) before moving on?

Methods: There needs to be more contextual and methodological detail around the process of recruitment and data collection, for the readers’ clarity. It is not clear for example, how and where participants were approached and recruited, where the data was collected and under what arrangements. Furthermore, authors should also provide some detail on the development of the interview guide.

Results: The analysis is presented in sufficient detail elaborating all the identified themes. However, the authors frequently present a descriptive summary of their findings which is backed by an excerpt suggesting the same thing. For analytic depth, authors are recommended to keep this repetition to a minimum while also reducing the length of the texts used as excerpts.

Reviewer #2: This is an innovative and useful paper that explores multimorbidity in a qualitative manner using NPT and BOTT approaches. I found this use of these approaches relevant and interesting.

The literature review is comprehensive and the theoretical framework firm.

I would certainly like to see this paper published.

I have a few minor points to make:

First. I thought the number of people interviewed could have been slightly larger than 32, perhaps 40. However, the authors have acknowledged the limitation of their work and the current number of 32 will do.

Second. The paper suffered from wordiness in some sections; for example, and particularly, at Mobilising capacity 1.2. This section and others could do with a little tightening to reduce the wordiness.

Third. Ufa woyera is not 'fermented' but corn or maize floor made from de-husked corn/maize.

Reviewer #3: I think this study represents a welcome addition to the literature, studying the experiences of patients with multimorbidity in Sub-Saharan Africa. It is a well performed qualitative study that must have brought many challenges for successful completion, for which I would like to compliment the authors.

The paper could be improved by a very precise statement of the goal(s) of the study, and then reporting on the methods and results for all goals set, and finally a reflection on their meaning and a compatible conclusion.

From the Abstract and Introduction, it’s clear to me that investigating the experiences of Malawi people with multimorbidity is an important study goal. However, it’s not clear to me whether or not “investigating their treatment burden of multimorbidity” is a study goal on itself. There is a clear focus though on the multimorbidity ‘burden’ throughout the paper.

The Introduction states that “investigating the usefulness of NPT and BoTT for structuring the analytic accounts of these patients’ experiences” is another goal of this study. I find this goal hard to grasp. Unfortunately, the following parts of the paper do not succeed in explaining (to me) the use of this study goal (what does this research question mean and why is it a relevant question?), the way how exactly the researchers investigated this question, neither in what it resulted.

If studying “the burden of multimorbidity” is aimed, I think it makes sense to use these instruments that have already shown their value, although in different settings. It seems to me that the researchers had good reasons to study “the multimorbidity burden” in Malawi, aided by NPT and BoTT, to assess similarities and differences in these patients’ experienced “burdens” as compared to patients with multimorbidity in non-SSA countries.

The discussion of the paper needs some improvement / rewriting. Generally, this section of the paper starts with a summary (what again was the study goal, which are the results we found). The paragraph that is meant to compare the findings to existing literature, reveals to me an important novelty, a strength of the current study, namely the assessed importance of social networks and resources affecting the multimorbidity burden as, now for the first time, assessed in a SSA country, by applying the theories of NPT and BoTT, and revealing a new awareness, namely the burden of ‘lack of treatment’. The authors may consider writing an ‘implications for practice (and science)’ paragraph and label it accordingly. The strengths and limitations section is not highly reflexive and inaccurately written (it states that 4 conditions were included – the methods state only 3). Some items discussed in lines 214-224 could be dealt with here. When reverting to the exact research question(s), also the conclusion paragraph could be somewhat more concise and crisp. And it could correspond somewhat better with the conclusion as stated in the abstract.

In the results section, the citations do not always clearly illustrate that what is stated by the authors. For example: line 538 and the citation following.

Line 551 and the following citation is another example where the citation gives quite general statements and does not provide tangible ‘solutions’.

The paper might benefit from comments from the authors on non-verbal communication (if studied / recorded). Reflection on cultural aspects (in the discussion) might be helpful for western readers of this study. For example, important factors when interviewees comment on organisations and persons with authority such as doctors, governments. Or on the communication between interviewer and interviewee, e.g. if inconsistencies would be noticed (money might enable patients but doesn’t immediately solve the issue of unavailability).

Minor comments:

Line 222-223: comments such as these should be more appropriately placed elsewhere (not in the Methods section).

Line 242: I see it as a respectful and appropriate step to mention that community leaders were asked permission (Ethics paragraph).

Line 528-532: reading this quotation, labelled as ‘choosing care providers’ and concerning ‘stigma’, to me the question arises whether ‘low trust in, or concerns on doctor’s professionality’ (disclosing confidential patient information to others) could be seen as a (recurrent) theme, maybe also in other interviews? I wonder what the author’s view is on this issue and how it impacts (or doesn’t impact) the burden of multimorbidity.

The abstract contains a duplication in the results. It could focus more on the new findings in Malawi, compared to already known aspects defining the multimorbidity burden from previous studies (in western countries).

6. PLOS authors have the option to publish the peer review history of their article (what does this mean?). If published, this will include your full peer review and any attached files.

**Do you want your identity to be public for this peer review?** For information about this choice, including consent withdrawal, please see our Privacy Policy.

Reviewer #1: **Yes: **Faraz Siddiqui

Reviewer #2: No

Reviewer #3: No

---

## [Decision Letter · Decision Letter 1]

1 Feb 2022

Experiences of multimorbidity in urban and rural Malawi: an interview study of burdens of treatment and lack of treatment

PGPH-D-21-00403R1

Dear Dr Bunn,

We are pleased to inform you that your manuscript 'Experiences of multimorbidity in urban and rural Malawi: an interview study of burdens of treatment and lack of treatment' has been provisionally accepted for publication in PLOS Global Public Health.

Best regards,

Melanie Boeckmann

Academic Editor

Dear Prof. Bunn

thank you very much for your patience and your thorough revisions!

All the best

Melanie Boeckmann

Reviewer Comments (if any, and for reference):

Reviewer's Responses to Questions

**Comments to the Author**

1. If the authors have adequately addressed your comments raised in a previous round of review and you feel that this manuscript is now acceptable for publication, you may indicate that here to bypass the “Comments to the Author” section, enter your conflict of interest statement in the “Confidential to Editor” section, and submit your "Accept" recommendation.

Reviewer #1: All comments have been addressed

Reviewer #2: All comments have been addressed

Reviewer #3: All comments have been addressed

2. Does this manuscript meet PLOS Global Public Health’s publication criteria? Is the manuscript technically sound, and do the data support the conclusions? The manuscript must describe methodologically and ethically rigorous research with conclusions that are appropriately drawn based on the data presented.

Reviewer #1: Yes

Reviewer #2: Yes

Reviewer #3: Yes

3. Has the statistical analysis been performed appropriately and rigorously?

Reviewer #1: Yes

Reviewer #2: N/A

Reviewer #3: N/A

4. Have the authors made all data underlying the findings in their manuscript fully available (please refer to the Data Availability Statement at the start of the manuscript PDF file)?

Reviewer #1: Yes

Reviewer #2: Yes

Reviewer #3: Yes

5. Is the manuscript presented in an intelligible fashion and written in standard English?

Reviewer #1: Yes

Reviewer #2: Yes

Reviewer #3: Yes

6. Review Comments to the Author

Reviewer #1: n/a

Reviewer #2: I am happy with the paper as it is.

Reviewer #3: I am happy to accept the changes made by the authors. My compliments for the valuable contribution to international multimorbidity literature they make once this paper gets published.

7. PLOS authors have the option to publish the peer review history of their article (what does this mean?). If published, this will include your full peer review and any attached files.

**Do you want your identity to be public for this peer review?** For information about this choice, including consent withdrawal, please see our Privacy Policy.

Reviewer #1: No

Reviewer #2: **Yes: **Dr John LWANDA

Reviewer #3: **Yes: **Hilde Luijks
